# The Historical Small Smart City Protocol (HISMACITY): Toward an Intelligent Tool Using Geo Big Data for the Sustainable Management of Minor Historical Assets

**Valentina Pica** [1,*], **Alessandro Cecili** [2], **Stefania Annicchiarico** [2] **and Elena Volkova** [2]

[1] Department of Architecture, Roma Tre University, Rome 00184, Italy
[2] Department of Sciences, GIS Laboratory, Roma Tre University, Rome 00184, Italy;
   alessandro.cecili@uniroma3.it (A.C.); stefania.annicchiarico@uniroma3.it (S.A.);
   elena.volkova@uniroma3.it (E.V.)
**\*** Correspondence: valentina.pica@uniroma3.it

**Abstract:** This research reports the ongoing design of the HISMACITY (Historical Small Smart City) Protocol, a planning tool with a certification system. The tool is designed for small municipalities in Europe. Through the award-winning certification system, the Protocol supports the fulfillment of best practices. Such practices can enhance town attractiveness. It also counteracts excessive land use that results from urban growth, and reduces demographic decline in internal areas of each country. The research methodology is grounded on building a dynamic dataset using geo big data, local data, and mobile data via information communications technology (ICT), and real-time data through sensors. The tool aims to build algorithms to calculate indicators that measure quality standards of integrated interventions. The aim is to reach specific goals within defined priority areas of the Historical Small Smart City Protocol. Being highly adaptive, the framework follows urban responsive design principles based on weighted suitability models that can be calibrated by changing the input data and the weights of the linear combination formula. The results highlight varying framework data, including the tool's development procedures and practicality.

**Dataset:** Link to the deposited dataset: https://zenodo.org/record/2547851#.XEj-BVxKhPY. Title: HISMACITY pro Dataset Framework; DOI:10.5281/zenodo.2547850. Link to the participatory planning map and current preliminary results https://www.arcgis.com/apps/MapSeries/index.html?appid=8870d0be46bc46acad6de2066e1e1a4b.

**Keywords:** Smart City; ICT; remote sensing; GIS dataset framework; geo big data; urban responsive design; valuation; indicators; weighted suitability models

---

## 1. Summary

The Historical Small Smart City Protocol (HISMACITY) is a dataset framework based on GIS (geographic information system) software that is primarily implemented using open big data and local data. Its purpose is to assess future scenarios for developing integrated strategic planning that is oriented toward sustainable management, in order to develop and preserve minor historical centers [1]. The necessary data types require the use of GIS tools for correct standardization, homogenization, and classification, as the data can then be catalogued into hierarchical datasets

and analysis models. The project defines different levels of interaction for this information using a single analytical framework for calculating the indicators that support the Protocol's evaluation criteria. This study was funded by the Horizon 2020 "Nurturing Excellence by Means of Cross-border and Cross-sector Mobility" (EU.1.3.2.) of the Marie Sklodowska-Curie Actions (MSCA) Individual Fellowship program [2].

Today, sustainability must be linked to the intelligent, focused use of technology to optimize governance tools. Traditional governance plans for land use have proven not only rigid and ineffective, but also unable to adapt to complexities arising from globalization, excessive land consumption, pollution, technological revolution, and new productive processes. Locally, municipal urban plans are usually prescriptive and static, and thus exclude the dynamic possibilities of shared initiatives. The use of new instruments is therefore necessary [3]. The situation is now critical, as a result of postmodern transitions compounded with climate change. In order to address these issues, it is clear that every new planning tool should now include civil society participation within public administration initiatives, for the purpose of influencing effective urban policies [4]. Among such factors, the Protocol's dynamic framework can affect various municipal and area-wide policies. This includes reviving local communities' sense of belonging. The Protocol also touches upon environmental protection and the management of urban common goods in the historic and cultural landscape, including tangible and intangible cultural heritage. The Framework tries to integrate all of these aspects, linking them to sustainable development management and disaster risk management [5]. Therefore, it involves various fields of action and disciplinary areas.

The Framework's main goal is to reactivate settlement dynamics in minor centers. Most settlements are located in "internal" or rural areas. The subsequent physical revitalization and safety of the built and naturalistic environment is also part of this goal. This can be applied to Italy, as well as other countries in Central and Southern Europe, where slowed economies are common [6]. Overall, the potential benefits of publicly releasing the dataset content include contributing to government systems that help to facilitate and place cities and their territories into integrated intervention programs, with a broad consensus and optimized use of funds. This new scenario could provide extensive possibilities for sustainable development, including cultural heritage valorization and protection.

The possibility of using certification systems supported by digital technology innovations, such as HISMACITY, may reduce the time required for this process, allowing it to become a manageable and measurable variable for monitoring progress with the possibility of making actions more efficient and economically sustainable. Projects using Industry 4.0 technologies could allow data coming from the external environment to interact with their own structures to achieve and maintain predefined objectives and/or different performance parameters. Through the use of the various developments of Industry 4.0 technology, such as the Internet of Things (IoT), Information and Communications Technology (ICT), cloud computing, analytics, augmented reality (AR), virtual reality (VR), and mixed reality (MR), it is now possible to build useful tools for urban regeneration project management. Such tools support an interaction among public administrations, citizens, and various actors in the territory.

The project also aims to pursue the following specific goals identified through a literature review on "smart cities":

1. Elevate the attractiveness and quality of life in minor centers by supporting the improvement of basic services, public space security, citizen services provided by the public administration, and cultural and recreational initiatives.

2. Increase economic competitiveness through the marketing of local products. Small cities have always been centers for the exchange of goods. Nowadays, small cities can even promote the production and sale of local products, as well as specialized services for culture and tertiary sectors. They can support the creation or growth of short supply chains, with effects on productivity and

employment as well as the opening of new sectors and vertical markets supported by 5G technology (i.e., e-commerce, virtual reality, e-marketing).

3. Ensure sustainability in the strategic planning of integrated interventions. Economic growth and quality of life must be promoted under a sustainability viewpoint, in order to guarantee a legacy for future generations. Part of the Smart City vision is the promotion of ecological and efficient use of natural resources and renewable energy [7].

The architecture of this technological tool will be released with its content at the project's close in 2019. However, the project aims to be a constant work in progress that is open to new implementation and feedback from both territorial and institutional bodies.

## 2. State-Of-The-Art and Data Description

Today, urban-scale application trials of technological tools for public space monitoring and integrated management, together with the use of experimental tools for the maintenance and recovery of abandoned or green areas and historic or registered buildings, are almost non-existent. Nevertheless, various sectoral projects are being carried out all over the world for cultural heritage management and the mitigation of hazards, through the use of new technologies, both in the entrepreneurial field and through experimental research [8].

The use of sensors associated with georeferenced systems is one of the most widespread fields in this area. In the sector of cultural heritage tourism, this, along with virtual architecture and augmented reality, is linked to reconstruction following natural disasters or the development of museological models [9]. Despite the now extensive availability of even low-cost tools, there is still a lack of software platforms that permit linkage with monitoring systems or the use of spatial data models for the analysis and simulation of possible scenarios. Various aspects of urban planning are targeted at the recovery of environmental beauty, as well as the security and protection of the landscape.

In order to allow these systems to create and realize integrated urban sustainability projects, and to achieve the goals dictated by international standards, like the United Nations 2030 Agenda, the development of broadband and ultra-broadband is essential, even though it has not fully reached Europe. The use of broadband and ultra-broadband will lead to the use of 5G technology. This will be made possible via the use of fiber-optic infrastructure and retrofitting buildings that include multi-service systems. In Italy, for example, the Financial Planning Association of Australia (FPA) 2017 report [10] shows that the country is halfway towards its goal of connecting to 30 Mbps speed (right now, reaching only 55.9% of the connected population), and far from its digital agenda for services goal of connecting to 100 Mbps, which today reaches only 10.7% of the total connected population. At the top of the Italian "smart cities" ranking from the same FPA report, are cities that have been able to combine energy transition, risk mitigation, and ecological conversion by supporting this process with advanced and mature planning, as well as governmental tools.

### 2.1. The HISMACITY Dataset Framework

The HISMACITY Dataset Framework is a certification system targeted toward municipal councils that was designed to allow the automatic calculation of indicators of possible scenarios of sustainable development. It also provides clear guidelines for the achievement of defined objectives in the field of urban planning. Beyond that, the tool can score the interventions included in the tables (which are organized by category of action) on the six dimensions of the Historical Small Smart City Protocol (mobility, economy, environment, heritage, living, and governance). These categories, or priority areas, were identified and designed following preliminary research in the area, which included a review of the literature and projects on environmental sustainability and smart city conceptual models [11].

The process of building the Framework extends from the data model concept to the creation of its architecture through the organization of different kinds of data. Input data are used to create algorithms for the calculation of specific indicators associated with the evaluation of threshold criteria that are linked to defined goals. This framework also allows dynamic use of real-time data obtained

from drones, control units, and sensors to return output data to the user that can be used to monitor the current status and asses critical issues. This is useful for achieving specific performance standards.

### 2.1.1. Different Components of the HISMACITY Protocol

The operative system of the Framework is a portal hosted on a WebGIS platform. The use of this portal is an initial experiment prior to the construction of an open source platform that can optimize the management of minor historical centers. This portal will support policies on integrated sustainable development and remote monitoring for security and risk management. The developed guidelines, together with the platform, constitute the technical tools of the certification system which will provide the scores for various interventions.

The Protocol is therefore a parametric group of instruments that supports policy decisions, i.e., a DSS (decision support system) [12]. It will provide a dynamic foundation for ensuring the applicability and replicability of its certification system. It will follow responsive urban design criteria, because it will be possible to modify the input data and correct the evaluation process progressively by adapting computed algorithms to any specific local needs [13] (Figure 1).

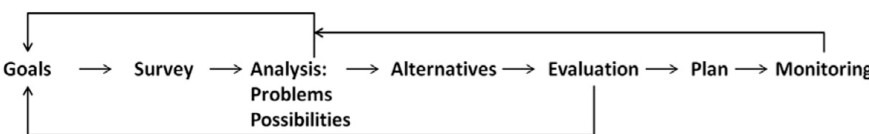

**Figure 1.** Responsive design approach and advanced alternative selection.

Adaptability and scalability are achieved by modifying the weight values in the linear combination formula of the weighted suitability models, as detailed in the "Methods" subsection.

The guidelines for Italy's most utilized certification system on sustainability in construction, the ITACA (Institute for Innovation and Procurement Transparency and Environmental Compatibility) Protocol, are associated with evaluation software that was originally developed by Excel, produced by Microsoft. This uses a methodological approach and allows the system to be used through an easily understood and non-geographic interface. In contrast, the HISMACITY Protocol can also be used in association with the WebGIS platform to provide an environment that is able to take advantage of the open source applications of the online Web Server ArcGIS, such as calculation models and vertical applications (see Appendix A).

To summarize, the Protocol's certification system will consist of three elements: a geographic portal for data analysis and evaluation, tables with evaluation criteria linked to the assigned bonuses, and the guidelines document (Figure 2).

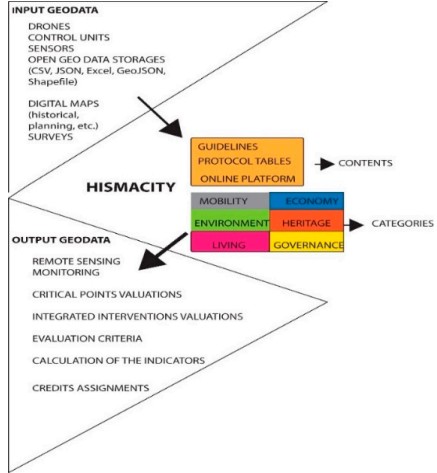

**Figure 2.** Structure and content of the Historical Small Smart City Protocol (HISMACITY) dataset framework.

### 2.1.2. Open Data Usage in the HISMACITY Dataset Framework

Of the big data available today, the system uses open geo data for the analysis, allowing users to preconfigure future concrete scenarios and share information in real-time through open source systems in cloud database repositories. Open geo data published freely online by municipal, regional, and national authorities made up the bulk of what was used to build the Framework. They were processed by extracting the necessary content in various standard formats, including CSV, JSON, Excel, GeoJSON, and Shapefile.

The data were processed individually and shared online via the portal server to allow the users responsible for managing the system to view and interact with them.

The development process of the HISMACITY Dataset Framework started with the homogenization of data, followed by the creation of a relational database, and finally, the production of complex spatial analysis models for the identification of environmental indicators that support the evaluation criteria. The Framework was designed to allow users to read and question cartographic data and to provide summary tables derived from the calculations. Thus, users can participate in an online observatory to detect critical issues (Figure 3).

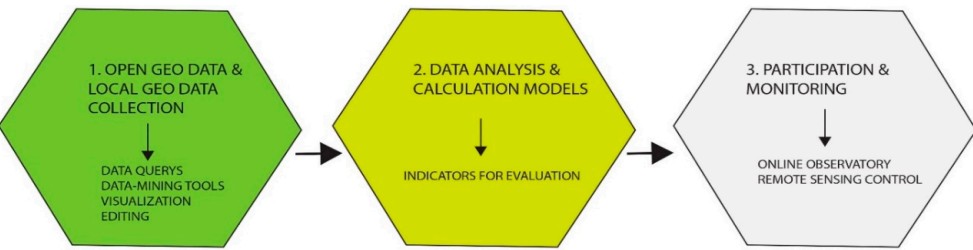

**Figure 3.** The HISMACITY development process and practicality.

Thus far, the calculation results of the first indicators and the dynamic story-map for the planning process have been published. These aspects can be used to complete the preliminary analysis and establish a methodology to validate the evaluation criteria.

The implementation of data onto a dynamic platform to support political decisions to form HISMACITY is dependent on the municipal administrations and their technical offices, which should receive increased capacity or support to carry out such tasks, as well as guidance regarding the retrieval of sensitive data (e.g., from social survey campaigns) and/or big data.

### 2.1.3. The HISMACITY Web Map for Participatory Process

The participatory process helps to build and verify the evaluation criteria of the Protocol by allowing people access to an open source dynamic map, which is included in the WebGIS platform, during the formulation of its content, as well as during a subsequent validation phase. The use of ICT and its relevance in the Framework design is observed through this operation, as users can access the web map from mobile phones and tablets. The need to valorize specific local strengths and priorities can hopefully be highlighted through social interactions. The evaluation and validation of the Protocol can also be carried out progressively, following the "Open Cities" criteria [14], and by leveraging other existing open source tools, as well as crowdsourcing and open data platforms. Secondly, the Protocol supports the creation of living labs, open labs, or urban labs for new, functional, and productive ideas. A number of more significant unused or misused buildings may be identified as having the potential to be refunctionalized. The locations of the buildings could be clearly shown in relation to well-known tourist sites, in order to draw tourists to them.

### 2.1.4. The HISMACITY Contribution to Administrative Streamlining

All of the aforementioned uses could significantly facilitate the management of public affairs by local administrations. This management involves the services provided to citizens (e-governance),

as well as maintenance and planning. It is valid even for observing European directives, which have led public bodies to move gradually towards widespread data distribution to the general public (citizens, professionals, public, and/or private bodies).

The Framework will provide a significant advantage in terms of increasing the quality of spatial analysis, even for replicability purposes. Suffice it to say, the identification of environmental indices requires algorithms involving various data coming from different sources, which are difficult to obtain by using other software. Thanks to GIS tools, it is possible to create and apply spatial analysis models that can be replicated under different territorial contexts by varying only the input data.

The Protocol's open source distribution will allow local public administrations to use the results of the research. They will be able to consult the tables and guidelines together with data input and the results of the analysis, which will be fully released online by the ArcGIS Online webmap. Additionally, the complete dataset framework will be available on this web server.

### 2.1.5. The Design Phases of the HISMACITY Dataset Framework

The development of HISMACITY is currently proceeding according to the following research and design phases:

1. Data is organized in order to analyze the current state of the minor historical centers where the Protocol's parameters could be applied, as well as the use of calculation models to set indicators. The pilot study is the Sutri project in Italy's Lazio region, in the province of Viterbo.

2. Geo big data is collected and integrated with local and sensitive data, mainly retrieved from in situ surveys. Local data are not only obtained through onsite inspections, but also through interactions with the municipality's technical office. As they are often in DWG format, they need to be georeferenced and homogenized. The geo big data are released by portals in the Lazio region, as well as the National Institute of Statistics (ISTAT), the National Institute of Economic Activities (ATECO), and other institutional bodies.

3. Integration and critical analysis of data is carried out to create descriptive frameworks to summarize the critical points and potential opportunities for the analyzed historical center.

4. Scenarios are evaluated and indicators are constructed for the measurement of priorities and intervention capacities related to score allocation in the Protocol tables, i.e., higher values of the beneficial indicators will be associated with higher scores in the Protocol and higher values of the disadvantaged indicators (related to the degree of road danger or demographic decrease) will be associated with lower scores connected to those specific indicators in the Protocol.

5. Possible projects are summarized with assigned intervention priorities. These are identified from the indicators related to the various priority areas or categories of action. In this phase, interventions for urban regeneration and historic building recovery with energy efficiency are included in the GIS system.

### 2.2. Dataset Framework Description of the Contents and the Procedures

Currently, only data on the Sutri municipality have been included in the Framework. The data analysis is being carried out according to two procedures. The first is input data collection, which is achieved using both geo big data and local data, which are organized in a dataset that details the current state of the Sutri site. The analysis mostly relates to the historical center on an urban scale, although some data describe the municipal territory and province, as these are useful to define indicators that connect the historical center to the inter-communal context (e.g., the Local Public Transport Efficiency Indicator).

The second phase of the analysis involves the reorganization of data in order to build models to calculate the evaluation criteria indicators for the six dimensions of the Historical Small Smart City Protocol.

In the first phase, the dataset is ordered by subfolders using geodatabases that identify the four planning systems referred to in the existing provincial plans, particularly the PTPG (General Provincial Territorial Plan) of Viterbo. These geodatabases include:

1.  A settlement system,
2.  An infrastructure system,
3.  A landscape-cultural system and,
4.  A socio-economic system.

The local data are cataloged in the same repositories as the geo big data, but within specific subfolders.

### 2.2.1. Data Input Collection Phase

The data collection phase involves the normalization and homogenization of data from different sources, giving them single vector and raster formats, as well as a single reference system. Thus, feature classes are obtained within each geo database.

The feature classes included in the aforementioned geodatabases are referred to in the analysis of each planning system. They are described in Table 1.

**Table 1.** Feature classes included in the geodatabases of the current state analysis dataset.

| Geodatabases | Geo Big Data | Local Data |
| --- | --- | --- |
| Settlement system | • Provincial framework from the regional technical map [15], scale 1:5000<br>• Historical centers from PTPR (Regional Landscape Territorial Plan) in the open data Lazio portal and municipal areas from the regional technical map [16], scale 1:5000<br>• Updated cadastral cartography of the municipalities from the regional technical map [17], scale 1:5000<br>• Zoning and classification using the municipal plan and Italian Ministerial Decree 1444/68 [18] | • Building usability (permanent, temporary, unused, mixed)<br>• Usability of open spaces (public roads, rest areas, public parking)<br>• Usability of unused spaces (ruins, closed cellars) |
| Infrastructure system | • Energy networks (from PTPG) [19]<br>• Railway networks (from PTPG) [20]<br>• Roads in the Lazio region (Open Street Map) [21]<br>• Commuting flow from Sutri to the municipalities of the inter-municipal territory by bus or private car (ISTAT data) [22] | • Tourist routes linked to Via Francigena<br>• Trekking and mountain bike trails<br>• Parking lots and interchange points<br>• Bus stops |
| Landscape-cultural system | • Hydrogeological plan (PAI) of the Province of Viterbo (Ministry of Environment, web feature service) [23]<br>• Geolithological map of the Province of Viterbo (open data Lazio portal) [24]<br>• Hydrographic constraints (Ministry of Environment [25] and open data Lazio portal [26]) with buffer zones | • Attractions: museums; urban walls; parks; archaeological sites; churches and sanctuaries; historic streets; period-specific historic buildings<br>• Municipality-owned buildings<br>• Public squares<br>• Degradation levels (high, low, medium, ruins)<br>• Building consistency: classification of minor buildings and monuments in the historical center<br>• Coverage types of minor buildings and monuments in the historical center<br>• Existing functions on various levels: organized from the basement up to the sixth floor<br>• Building types<br>• Types of open space (connection/relation spaces, streets, squares)<br>• Cadastral comparison of the historical land registers with the current land registry [a]<br>• Overhangs/additions to be removed |

**Table 1.** *Cont.*

| Geodatabases | Geo Big Data | Local Data |
|---|---|---|
| Socio-economic system | • Agricultural activities (from data requested by the Viterbo Chamber of Commerce) [b]<br>• Breeding activities (from data requested by the Viterbo Chamber of Commerce)<br>• Corinne Land Cover (with soil classification: natural, agricultural, artificial, wetlands, bodies of water) [27]<br>• Agricultural system (from the Corinne Land Cover, to be interpolated with agricultural activities)<br>• Tourist accommodation (from Viterbo Chamber of Commerce data)<br>• Basic services: schools, health care services (from the open data Lazio portal [28]) | |

[a] Obtained by specific request from the Sutri Municipal Historical Archive. [b] Data requested to the Data Disclosure Manager of the Chamber of Commerce of Viterbo, Lanfranco Tenti lanfranco.tenti@vt.camcom.it.

The "building consistency" feature class included in the landscape-cultural geodatabase system has several classifications: churches; private property—special buildings; public property—special buildings; minor value and private property construction; minor value and public property construction; and undetectable. Each of these domains, visible on a table, is associated with certain characteristics, namely the number of underground floors, basements, and levels above ground as well as the types of unplastered material (reinforced concrete frames, perforated brick infill, load-bearing masonry in tufa blocks, solid brick load-bearing masonry, mixed masonry of tufa and bricks, mortar, plaster, contemporary brick curtains, and undetectable materials).

In addition to the aforementioned geodatabases, the georeferenced raster files on the enforced urban plans in the referenced municipal areas are inserted as layers of reading in a dedicated subfolder.

### 2.2.2. Data Analysis Second Phase

Once the phase of collecting and organizing the current state analysis data is complete, the data are reorganized into folders representing the six priority areas, or dimensions, of the Historical Small Smart City Protocol. The feature classes are entered into subfolders associated with new geodatabases, in order to identify the indicators in which the system should establish the computational algorithms.

To summarize, examples are fully described in the dataset related to the mobility field evaluation criteria, i.e., the first indicator of the first dimension of the Historical Small Smart City Protocol. However, having initiated the research only nine months ago, the full list of indicators is still in progress.

The current list is as follows:

• Economy: increase in touristic services, increase in local productivity, and energy saving;
• Environment: environmental monitoring, improvement in undifferentiated waste collection, and risk management;
• Heritage: efficient lighting, parking management, security, and smart buildings;
• Living: inclusion of ultra-wideband, efficient public WiFi connectivity, management and monitoring of internal traffic in historical centers, and local public transport efficiency;
• Governance: promotion of digital touristic services and creation of open access digital public services.

The geodatabases built for calculating the indicators of mobility are as follows:

1.    Pedestrian practicability;
2.    Pedestrian accessibility;
3.    Cycling accessibility;
4.    Securing open spaces and;
5.    Local public transport efficiency.

The feature class data included in the first geodatabase are as follows:
1.1.1.    Places of interest (from the landscape-cultural system geodatabase);
1.1.2.    Parking lots and interchange nodes (from the infrastructure system geodatabase);
1.1.3.    Basic services (from the socio-economic system geodatabase) an;
1.1.4.    Road graph from open street map (from the infrastructure system geodatabase).

The road graph is interpolated with the "slope" raster layer, which was created by a digital terrain model (DTM), and generated from the elevation points, with values available from the regional technical map of the open data Lazio portal.

Furthermore, the geodatabase dedicated to calculating the pedestrian practicability indicator contains:
1.1.5.    The centroid of the historical center;
1.1.6.    The border of the historical center;
1.1.7.    An isochrone of optimal accessibility (can be accessed in a maximum of 7 minutes or within 550 meters, in accordance with the European Community Urban Quality standards) [29] an;
1.1.8.    An isochrone of non-optimal practicability of distances of 1–1.5 km, covering all basic services in areas outside the historical center which are accessible on foot or by bike in a maximum time of 30 min.

The indicators come from the total weighted percentage of the pedestrian routes with non-optimal practicability and include points of interest, parking lots, interchange nodes, and basic services. The higher the non-optimal travel times, the higher the indicator's value.

## 3. Methods

It is anticipated that the Framework data's scalability will be validated in 2019, along with its analytical model, which will hopefully be applied to a minor Spanish center, such as Berga in the Province of Barcelona. The current dataset for calculating the indicators only concerns the pilot project for Sutri in Viterbo. The obtained shapefile of the "roads" feature class from the Lazio Region, which was retrieved via the open street map portal, has been processed to reduce the noise level by integrating the voids of missing streets or routes. Further, it has been clipped in order to adjust it to the targeted municipal area. Similar methods were followed in the processing of geo big data from open source portals. Moreover, even though the Protocol's tables and guidelines with the complete evaluation criteria include more indicators (Table 2), only the currently available local data and geo big data that specifically relate to the indicators mentioned, and that were created by the GIS model builder, were included in the dataset.

The table regarding the first priority area of action demonstrates simple quantitative indices that measure specific standards. The cycle path density in the mobility priority area is not included in the dataset, as it is predefined by the technical literature and is currently immeasurable in situ, e.g., using cycle lanes. The same methodological criteria will be observed to complete the datasets of the indicators included in the other five dimensions of the HISMACITY Dataset Framework.

Regarding the methodology for building the indicators' algorithms, the calculation of pedestrian accessibility is carried out within the second largest isochrone. The partial percentages of the practicability associated with each group of feature classes is given by the ratio between, and the sum of, the distances from the elements (places of interest, etc.) and the total number of pedestrian routes [30].

**Table 2.** Indicators and evaluation criteria for the mobility priority area of the Protocol that are not included in the dataset framework but only in the tables and guidelines.

| Evaluation Criteria | Indicators | Calculation Parameters |
| --- | --- | --- |
| Cycling practicability | Density of cycle paths | New or planned cycle paths (absolute value in km) |
| E-bike sharing services | Infrastructures with parking lots for e-bike sharing services | Number of e-bike sharing services and infrastructures |
| Minimum safety conditions of the cycle network and pedestrian paths | Cycle and pedestrian routes that are at least 3 m wide. | If they exist (yes/no), their security conditions, % km |
| Control and regulation of public and private mobility on wheels (remote sensing monitoring) | Impact of limited traffic areas on driveways in the historical center (total percentage value) | Weighted sum of the partial percentage values of Ztl (Limited traffic zone) areas/total area of the historical center driveways (percentage value) + n, remote monitoring devices to access the areas/n, total limited traffic areas (percentage) + number of hours of traffic limitation/total hours per day (percentage) |
| Electric bus lines | Electric bus cars for new or existing connection services | Number of electric buses/cars/km |
| Interchange points | Interchange points with specific quality standards | Number of interchange points/inhabitants |
| On-demand transportation | On-demand private transportation services | Number of cars/inhabitants |
| Info-mobility systems | Info-mobility systems to support local public transport and private mobility | Number of mobile apps for ticket purchases, sharing private vehicles, and service information |

The calculation model, which is a weighted suitability model, is based on the weighted sum of the partial percentages of practicability. This varies according to the trend given by the adequacy or efficiency level of the distance it takes to reach places of interest, parking lots, interchange points, and basic services by foot. These distances are included in the isochrone of non-optimal viability, and when they are particularly relevant, they can be corrected by the entry of sections of cycle lanes as well as private or public e-mobility services. Weights are assigned according to the value of the sites, i.e., greater historical/artistic relevance corresponds to a greater weight value. Therefore, places of interest (e.g., archaeological sites, artifacts, parks, and churches) are assigned a value of 10, while all other places are given a value of 5 in order to avoid discretion and arbitrage in evaluating relevance [31].

The data model's algorithm is better described through the following mathematical formula:

$$P_p = \sum_{j=m}^{n} (pj \times fj) \tag{1}$$

The formula indicates that the pedestrian practicability indicator ($P_p$) is a linear combination and can be solved by any software for linear programming problems. It shows that the final value is a total weighted percentage, resulting from the weighted sum of the values of the partial percentages (*fj*) multiplied by their weights (*pj*). The method used to assign the scores associated with the evaluation criteria involves the identification of quality standards for the interventions, which are set according to different performance levels. Each evaluation criterion is selected among those considered more highly effective for measuring the "smart"-ness of minor historical centers. This is in light of preliminary comparative research in the literature and other certification systems, like the Italian Green Building Council (GBC) *Quartieri* from the GBC Neighborhoods system [32]. The criteria must have a series of characteristics, borrowed from the ITACA Protocol:

- Significant economic, social, and environmental value;
- Quantifiable or qualitatively definable, or objectively responsive to predetermined performances;
- Broad target pursuit;
- Proven scientific value;
- Equipped with public interest prerogatives.

For each criterion, the municipal government receives a score that can vary from −1 to +5, which is assigned by comparing the calculated indicator with the general performance scale (benchmark) based on the previously defined indicators.

Zero represents the reference standard that must be considered as current practice in compliance with enforced laws or regulations.

The score assignment for each criterion is based on the indications and the verification method reported in the "descriptive paper" of each evaluation criterion in the guidelines document. In this document, each criterion is described using the following information:

1. Pursued need or quality objective;

2. Criterion weight representing the degree of importance assigned to the criterion with respect to the entire evaluation tool;

3. Performance indicator, which is the parameter used to assess the performance level of the historical center with respect to the evaluation criterion, it can be quantitative or qualitative and the latter is described in the form of scenarios;

4. Unit of measure, in the case of quantitative performance indicators;

5. Performance scale (or benchmark), i.e., the benchmark against which the performance indicator is compared for calculating the evaluation criterion score;

6. Verification method and tools that define the procedure for calculating the performance indicator of the evaluation criterion;

7. Input data, i.e., the list of data necessary for the calculation and/or verification of the performance indicator;

8. Documentation specifying the documents (or extracts) and sources from which the input data were extracted and contextualized;

9. Benchmarking, which specifies the methodology adopted for defining the benchmarks;

10. Legislative references, i.e., the enforced legislative provisions regarding the project;

11. Normative references, i.e., the technical reference standards used to determine the performance scales and the verification methods;

12. Technical literature, i.e., technical references used to determine the performance scales and verification methodologies.

All of the aforementioned references—criteria, benchmark, performance scale, indicators, units of measurement, and verification methods—are established by, and can only be modified by, the research team. The operative evaluation tool of the Protocol automatically updates the criteria and benchmark scales according to the type of project and specific local features of the historical center or territorial context.

The weighting system for the evaluation criteria is integrated with the weights assigned to the priority areas or the six dimensions of the Data Framework. They represent the degree of importance assumed within the entire evaluation system, with values assigned through "voting" and subsequent normalization of the assigned votes. The votes may vary within a range of 0 (area/category not applicable) to 3 (area/category with maximum importance). The assigned weights are established by the research team. They can be modified according to the future regional or national platform. The method is similar to what occurs in the application process of the Lazio region's ITACA Protocol.

The criteria's weights represent their assumed degrees of relevance. This includes two types: "relative", referring to the importance of a criterion within the priority area to which it belongs, and "absolute", which relates to its importance within the entire evaluation system.

Weights are assigned both to the criteria and the priority areas by estimating the environmental impact of each of them, assessed on the basis of three characteristics:

A—extension of the potential effect (3 = global or regional, 2 = urban or suburban, 1 = building or site);

B—intensity of the potential effect (3 = strong or direct, 2 = moderate or indirect, 1 = weak);

C—duration of the potential effect (3 = >50 years, 2 = >10 years, 1 = <10 years).

The subsequent normalization of the attributed votes is used to calculate the relative weight of each criterion. The absolute weight is the product of the criterion's relative weight in the weight of the priority area in which it belongs.

## 4. Ethical Issues

In compliance with the 1975 Declaration of Helsinki (revised 2003), in terms of the ethical issues involved in the use of sensitive data, this research includes two aspects regarding the analysis of data procured over the platform to be used for "peer to peer" urbanism and dissemination to a public audience: the right to protect personal data and the right to protect private residential information (e.g., planning, access, distribution). For this reason, the project obtained ethics committee approval prior to its undertaking. On 27 March 2017, the project was approved by the EU Commission department responsible for European Politics on Research and Development (Identification Code 743837). Undoubtedly, there is an ever-increasing pressure to safeguard the use of big data in "Smart Cities" creation to prevent breaches of information security, thus combining the exponential growth of technology with citizens' individual privacy protection. As privacy and information law experts Neil M. Richards and Jonathan E. King highlighted [33], meaningful privacy protection of big data can be achieved in the future through a combination of traditional regulations, "soft" regulations, and the development of big data ethics. This research aims to ensure privacy and security of data according to the Charter of Fundamental Rights of the European Union, as well as the European Convention on Human Rights and its supplementary protocols. These ethical aspects will be addressed by keeping these kinds of data and information absolutely confidential, with no sharing of data for secondary uses and no publication of any drawings of private domestic architecture. Such information will be used solely for research. Moreover, explicit approval or consent from participants will be required in all cases prior to collecting and processing data. The platform will also ensure the "right to be forgotten", so if an individual no longer wants their personal data to be processed and there is no legitimate reason for it to be kept by the research team, then it will be removed from the system.

**Author Contributions:** Project administration, investigation, original draft preparation, resources, methodology—V.P.; conceptualization, review, editing, formal analysis—A.C.; software, data curation, supervision—S.A.; webpage dataset, dynamic story-maps visualization—E.V.

**Funding:** This research was funded by the European Commission within the Horizon 2020 Program, MSCA-IF-2016 - Individual Fellowships, Register Number: 208560.

**Acknowledgments:** The authors acknowledge the administrative and technical support of Architect Mario Cerasoli in the Department of Architecture at Rome Tre University.

**Conflicts of Interest:** The authors declare no conflict of interest.

## Appendix A

Supplementary data to describe the WebGIS portal, which is available online, starting from the participatory observatory map and then showing, in succession, screens related to the input data organized by the four planning systems, current urban plans, and analyzed data in the category or field of action "mobility", including the practicability and accessibility indicators.

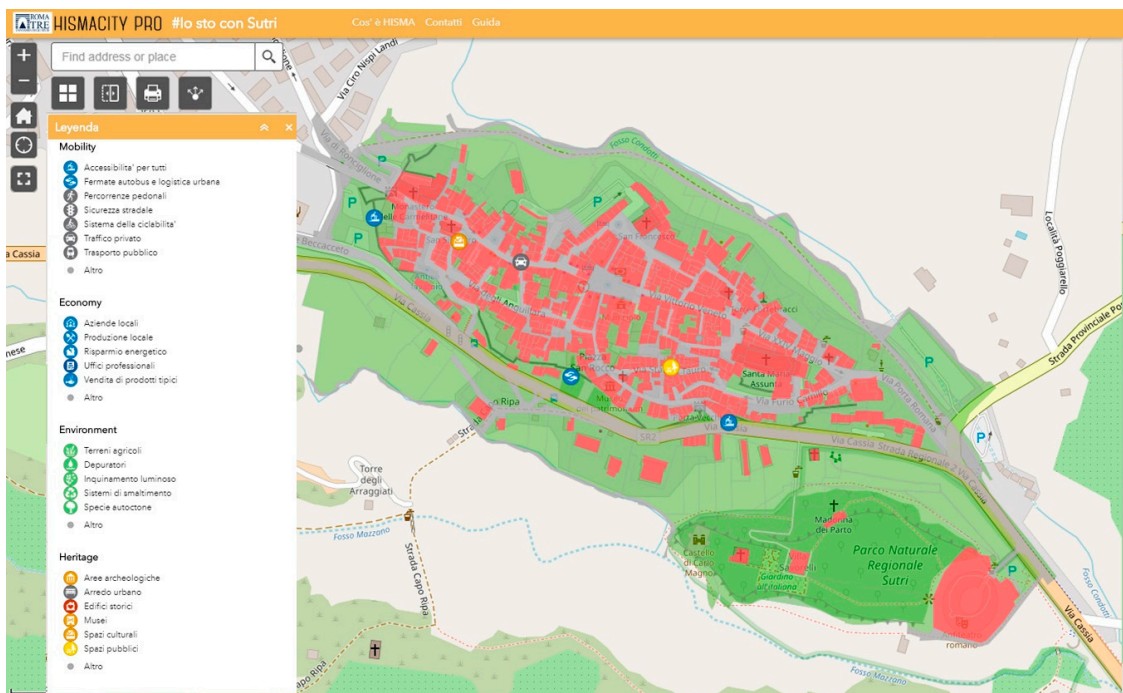

**Figure A1.** Webmap of the online observatory for the participatory process analysis of critical factors.

The legend of this map includes various indexes collected in the six dimensions of a small, intelligent, historical city that describe the possible critical aspects present in these fields. The icons that refer to these items are displayed on the map and are entered by users by clicking on the map and then selecting the reference field on the right.

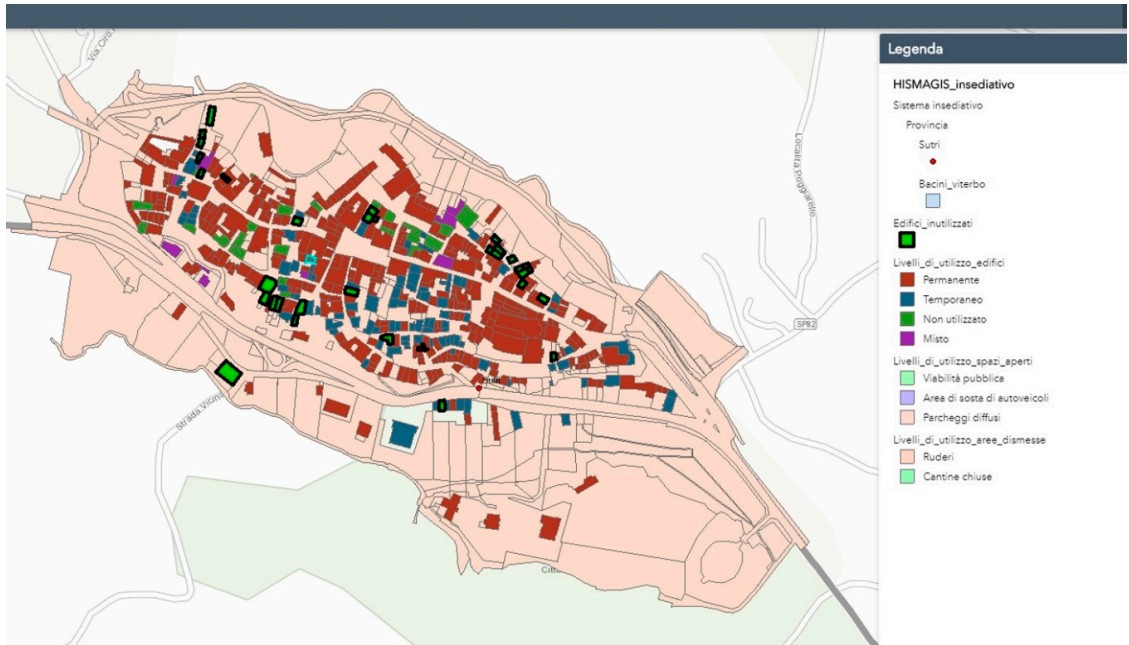

**Figure A2.** Webmap of the settlement system data, showing the usability level of the historic buildings in Sutri.

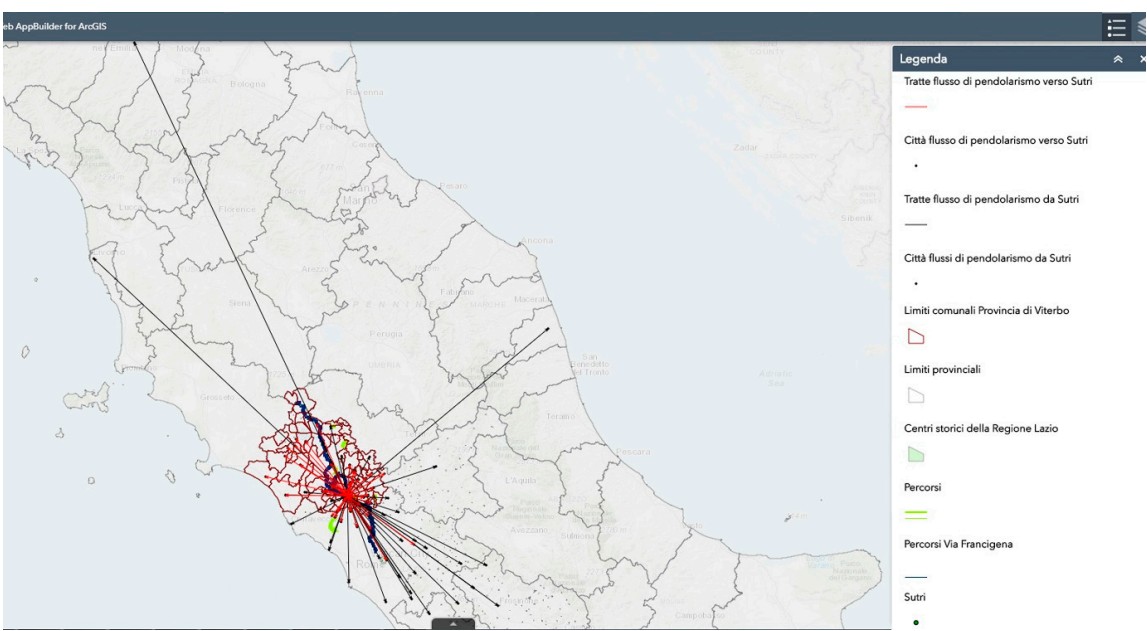

**Figure A3.** Webmap of the infrastructure system.

This map displays the rural and tourist routes, the rail networks, the major roads, and the commuting flow from Sutri to the inter-municipal territory.

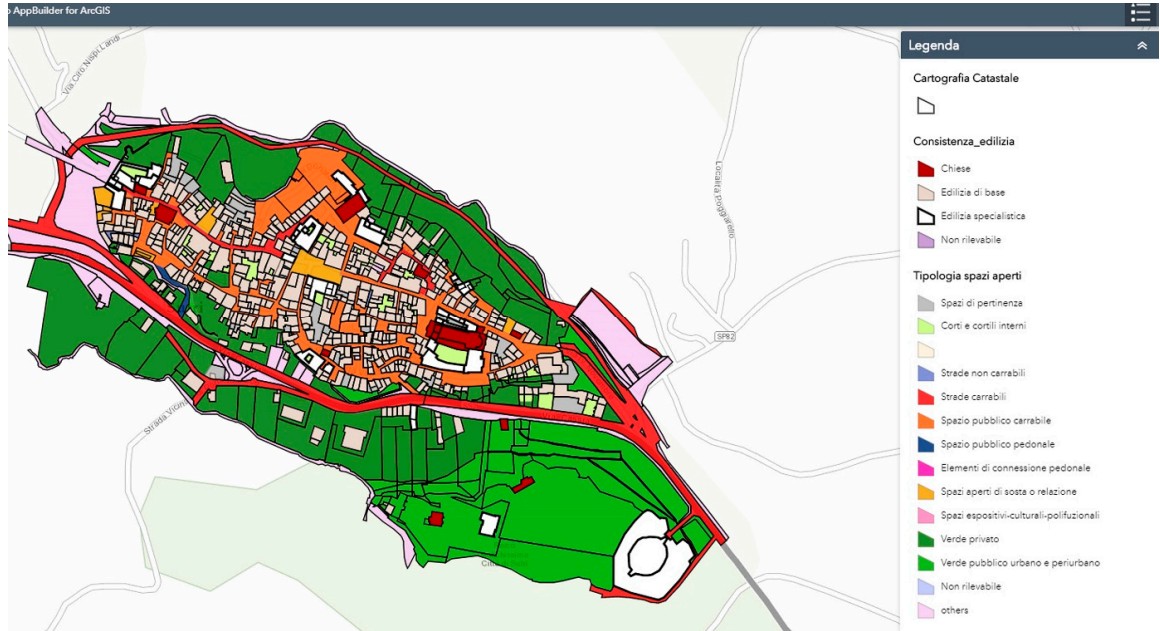

**Figure A4.** Webmap of the landscape-cultural system.

The figure displays the feature classes related to the architectural analysis of the quality of the urban environment, with the buildings classified as minor constructions ("edilizia di base") and special buildings ("edilizia specialistica"), such as historic palaces or monuments. Open spaces are also analyzed and divided according to various types (private green, squares, stairways).

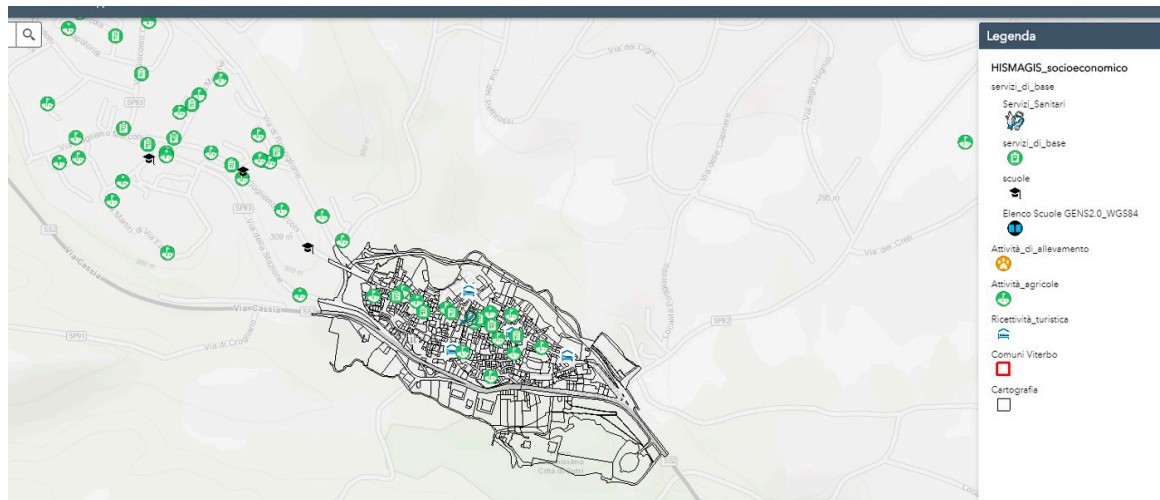

**Figure A5.** Webmap of the socio-economic system displaying the basic services, schools, farms, and tourist accommodations.

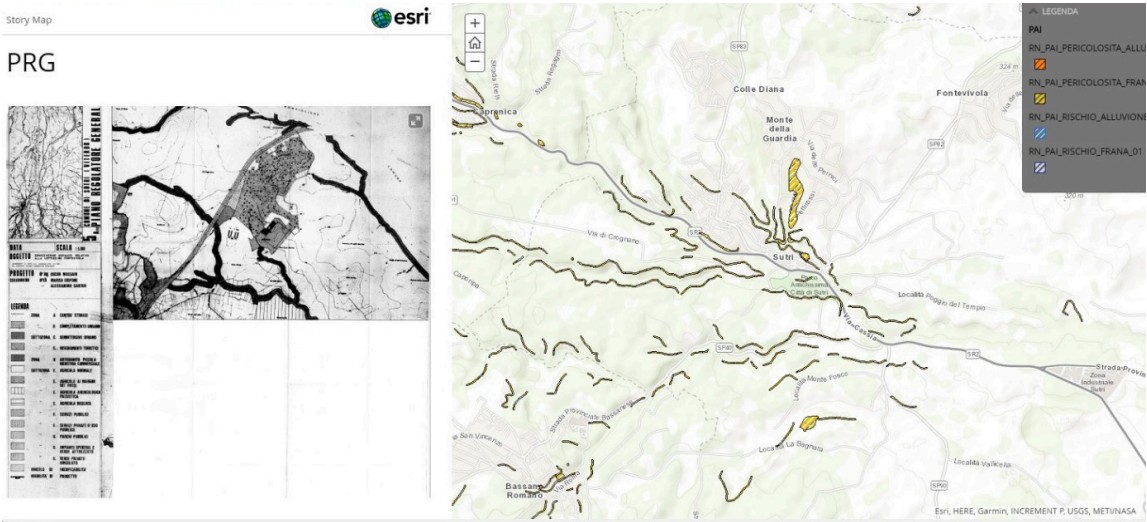

**Figure A6.** Webmap of the georeferenced raster files of the enforced urban plans in the referenced municipal area of Sutri, which are inserted as reading layers.

The image displays the General Regulatory Plan (on the left) and the Hydrogeological Structure Plan (on the right).

The figure displays, on the left, the calculation formula, and on the right, the different layers of the point of interest used in the formula.

The image shows the elements used in the calculation, indicated in the legend on the right as well as on the map. Among these, the buffer (area of delimitation of the optimal walking distance) was created with a GIS data analysis tool.

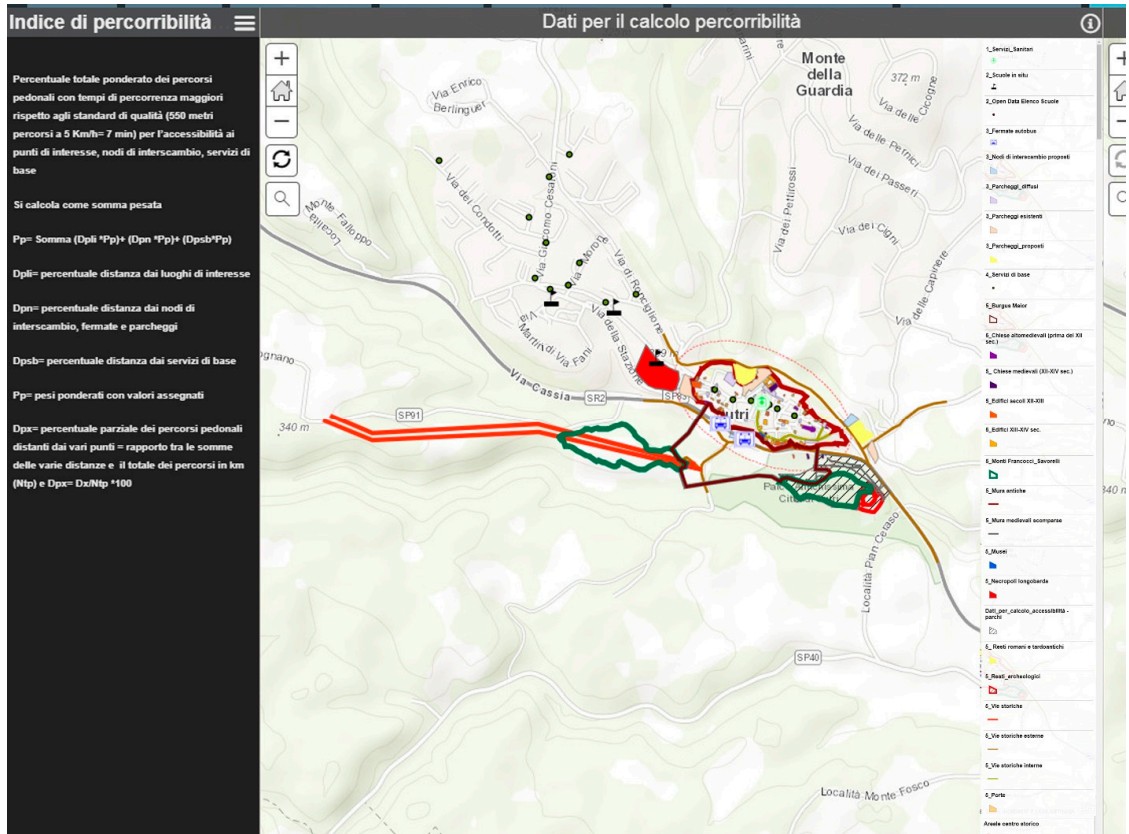

**Figure A7.** Webmap of the calculation of the indicators of pedestrian accessibility.

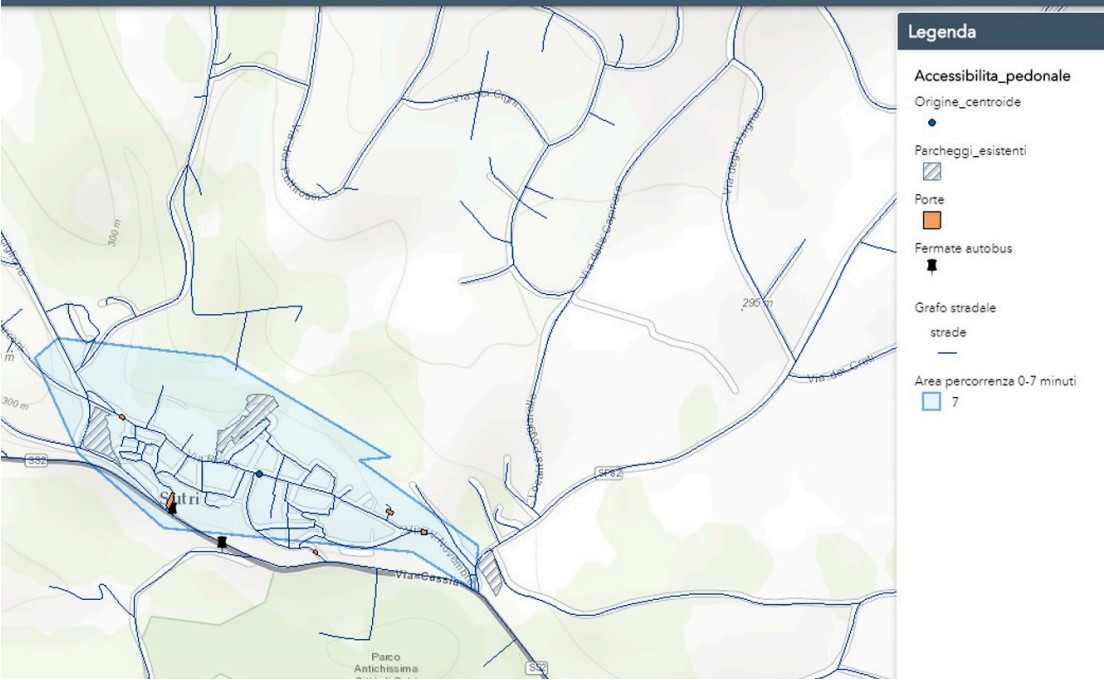

**Figure A8.** Webmap of the calculation of the indicators of pedestrian accessibility.

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
