# Peer review of "The Historical Small Smart City Protocol (HISMACITY): Toward an Intelligent Tool Using Geo Big Data for the Sustainable Management of Minor Historical Assets"

_data, 2018_

Reviewer 1 Report

The general idea of the project is interesting but the article is difficult to follow (looks more like a working paper).

The article is categorized as a Data Descriptor but I didn’t see any dataset attached, only a link to a map: https://www.arcgis.com/apps/MapSeries/index.html?appid=8870d0be46bc46acad6de2066e1e1a4b

It’s hard for me to apply the guideline https://www.mdpi.com/journal/data/guidelines without an actual dataset.

Author Response

The entire dataset is going to be downloaded on a web repository, probably Zenodo. The link is going to be attached in the section after the abstract in the paper.

The paper has been reorganized in subsections and english language corrections has been made by an expert.

The work tries to describe the dataset and explain its major outreach and expected results. in order not to make it appear has a working paper, some images of the published dataset have been included in the Appendix section.

Reviewer 2 Report

This manuscript presents a framework/protocol to support sustainable planning of historical small cities. In general, the idea is significant and meaningful, but the manuscript has a lot of potentials for improvements, which are explained below:

1 The structure of this paper needs to be improved. The current organization is very difficult to follow. I kindly suggest the authors to reorganize the manuscript in a clear scientific format.

2. Extensive editing of English writing is required. There are many informal expressions, typos, and redundancy in the current version. Some examples are listed below:

2.1 Lines 56-58: The sentence is very difficult to understand. Please consider revising the sentence.

2.2 Line 56: what are these factors? I could not find ‘these factors’ in the context.

2.3 Line 94: how 5G technology help raise the economic competitiveness?

2.4 Line 122: 5D or 5G?

2.5 Line 189: and or in?

3. One challenge of building a protocol or index-based frameworks is validation. The paper briefly mentioned the possibility to evaluate or validate the proposed protocol in lines 205 – 209. However, I kindly suggest the authors to list some examples of existing open source tools that can evaluate or validate the proposed protocol.

4. The arcgis app provides a lot of information, but the amount of work and outcomes are not well presented in the manuscript. It would be extremely helpful to provide some screenshots of the created dataset. In addition, the app is in Italian, so it might be better to provide English notes in the screenshots.

Author Response

1. The manuscript's contents have been reorganized in subsections.

 2. An extensive editing of English language and style has been realized on the paper by an expert.

2.1., 2.2.  The sentences 56-58 have been modified.

2.3. 94, now 87-88, some indications have been inserted.

2.4. now 116 line, has been modified.

2.5. now 207, modified.

3. the validation process is ging to be tested in the second phase, which is started since one month, and it is going to be investigated the possibility to use softwares of Multicriteria Analysis. Another way to validate it is through the open source platform which has been indicated and published in the Arcgis Server, by the interaction with the local community and administrators. the values introduced in the weighted sums of the indicator's calculations, for example, can be modified through this interaction.

4.Screenshots of the portal have been inserted in the Appendix A.

Reviewer 3 Report

This paper describes a geospatial big data integration, visualisation and analysis system. The methods and framework are well presented. The system is accessible for users. The paper might be improved from the following aspects.

First, statements should reorganised to reflect reasonable logics of contents. Subsections might be added within each section to clearly arrange statements. For instance, paragraphs in section 2.1.1 should be reorganised to several relatively long paragraphs to present the key ideas of this section. Meanwhile, English language should be improved.

Second, maps of primary outcomes in the system should be added in the manuscript to ensure authors' easy access to the data.

Finally, data sources should be clearly presented for each of the geographical attributes.

Minor comments:

Line 402 and equation 1: subscripts should be used for Pp, Pj and fj. 

Line 475 and 476: what are the meanings of two "B"s?

Author Response

The section has been modified, now having different subsections.

Maps hav been added in the Appendix A

Direct reference notes on the sources of the data of the georeferenced Geo Big Data part of the Table 1 have been inserted, and also for the cadastral comparison in the local data column of the table. All the other data proceed from direct surveys.

Minor comments: the two B have been modified (one B and one C, following). At the moment is not possible to modify the equation due to a word add on that is missing.

Round  2

Reviewer 1 Report

The dataset is now available for download. The methodology for data collection and normalization is presented. Maybe the potential interest of other researchers for the dataset could have been better stressed. Also, the English requires some extra polishing.

Author Response

The english has been further polished by an expert from the MDPI service.

Reviewer 2 Report

The authors have addressed all my comments and suggestions. The screenshots in the appendix are extremely helpful. One minor suggestion is that the scientific English could be improved. 

Author Response

(The authors gave the same response as above.)

Reviewer 3 Report

The comments have been addressed.

English language of this paper should be improved before publication.

Author Response

(The authors gave the same response as above.)
